# Quantifying the Influence of Surface Texture and Shape on Structure from Motion 3D Reconstructions

**DOI:** 10.3390/s23010178

**Published:** 2022-12-24

**Authors:** Mikkel Schou Nielsen, Ivan Nikolov, Emil Krog Kruse, Jørgen Garnæs, Claus Brøndgaard Madsen

**Affiliations:** 1Nano Research, Danish Fundamental Metrology, Kogle Allé 5, DK-2970 Hørsholm, Denmark; 2Department of Architecture, Design and Media Technology, Faculty of Science, Aalborg University, Rendsburggade 14, DK-9000 Aalborg, Denmark; 3AAU Innovation, Aalborg University, Thomas Manns Vej 25, DK-9220 Aalborg, Denmark

**Keywords:** photogrammetry, SfM, structure from motion, microscopy, power spectrum analysis, surface inspection

## Abstract

In general, optical methods for geometrical measurements are influenced by the surface properties of the examined object. In Structure from Motion (SfM), local variations in surface color or topography are necessary for detecting feature points for point-cloud triangulation. Thus, the level of contrast or texture is important for an accurate reconstruction. However, quantitative studies of the influence of surface texture on geometrical reconstruction are largely missing. This study tries to remedy that by investigating the influence of object texture levels on reconstruction accuracy using a set of reference artifacts. The artifacts are designed with well-defined surface geometries, and quantitative metrics are introduced to evaluate the lateral resolution, vertical geometric variation, and spatial–frequency information of the reconstructions. The influence of texture level is compared to variations in capturing range. For the SfM measurements, the ContextCapture software solution and a 50 Mpx DSLR camera are used. The findings are compared to results using calibrated optical microscopes. The results show that the proposed pipeline can be used for investigating the influence of texture on SfM reconstructions. The introduced metrics allow for a quantitative comparison of the reconstructions at varying texture levels and ranges. Both range and texture level are seen to affect the reconstructed geometries although in different ways. While an increase in range at a fixed focal length reduces the spatial resolution, an insufficient texture level causes an increased noise level and may introduce errors in the reconstruction. The artifacts are designed to be easily replicable, and by providing a step-by-step procedure of our testing and comparison methodology, we hope that other researchers will make use of the proposed testing pipeline.

## 1. Introduction

In recent years, Structure from Motion (SfM) has received increased interest. Aided by a rapid development of software solutions [1], SfM offer reconstructions of 3D geometric models in great detail with simple, fast, and low-cost acquisition [2,3,4]. As a result, potential applications of SfM for quantitative geometric measurements have been considered within a broader range of fields such as cultural heritage [5,6], geoscience [7,8,9,10,11,12], construction and architecture [13,14,15], and surface inspection [16,17,18,19,20,21]. Common for these fields is the need for accurate geometric models to allow reliable quantitative analyses. Since the applications cover large differences in object surfaces and capturing conditions, many factors may influence the SfM reconstructions. Among these are the local variations in surface color or surface roughness of the object. Both of these properties are often referred to as the ‘texture’ of the surface. While several recent studies have applied SfM for measuring surface roughness [10,11,12,22,23,24], quantitative studies of the influence of texture on the reconstruction accuracy are missing or provided for very specific use cases. This shows that further in-depth studies of the factors influencing the capture of micro- and macro-textures of 3D surfaces are needed.

In SfM, objects and surfaces are reproduced in 3D by using a number of image acquisitions from different distances and angles. The algorithms utilize image feature correspondences, sparse point cloud triangulation, and point interpolation to create dense point clouds and 3D meshes from the imaged objects and surfaces [25,26]. As the creation of the dense point cloud applies multi-view stereo algorithms, the full SfM pipeline can sometimes be referred to as SfM-MVS [26]. SfM has many advantages compared to other 3D reconstruction methods [27]. As it requires only conventional hardware such as a DSLR camera, SfM has relatively lower overhead costs compared to, e.g., laser scanning or LiDAR scanning. SfM can be used both indoors and outdoors [28] unlike structured light [29] and time of flight [30] scanners, which tend to fail when direct sunlight is present. Finally, it is capable of creating much more detailed reconstructions than most 3D-from-stereo systems. On the software side, a number of commercial SfM solutions are available. The work by [5] focuses on the low-cost and free solutions, in [31], incremental SfM pipelines with open source code are evaluated, while the work by [1,3,32] covers the newer free and paid software packages.

A drawback of SfM is that it is dependent on the quality of the acquisition system as well as on the physical properties of the captured object or surface. Since SfM depends on classical feature extraction, matching, and 3D triangulation, the angular coverage and resolution of the images greatly affect the quality of the reconstructed surface [33]. Thus, an increase in the number of images can improve the accuracy when additional angles and poses are added [4,34]. Another factor is the ground sampling distance (GSD) of the images. This depends on the focal length of the camera as well as the capturing range, which is the distance between the captured surface and the camera. A very large range gives a large GSD and can result in poor reconstructions [26]. As the spatial resolution of the acquired photos scales with range, the reconstruction accuracy scales accordingly [26,35,36]. Furthermore, the accuracy depends on the used optics and imaging equipment [37], the lighting conditions [38] as well as the surface properties of the captured objects. In SfM, the object surfaces ideally should not be glossy or transparent, and in the absence of external contrast sources, such as structured light, a level of texture on the surface is needed. This texture can either originate from a local variation in surface coloring or a local height variation of the surface topography, e.g., roughness. If too few distinct features are found on the object surface, this can result in a lot of false matches and degradation of performance [4,9] or outright empty regions and holes in the point cloud [39]. In contrast, other conventional 3D reconstruction methods are less sensitive toward surface texture. Laser-based methods such as LiDAR are not impacted by the local texture as long as a sufficient light intensity is reflected from the surface. In Photometric stereo first introduced by Woodham [40], light from different directions creates a shadow variation originating from the micro- and macro-surface details that is used to reconstruct the 3D surface [41,42]. While Photometric stereo can more easily capture smaller surface details, the limited light direction angles create difficulties in representing the larger overall surface shape. In addition, Photometric stereo introduces the need for an external light source that can move independently of the camera which would limit outdoor usage. Comparatively, SfM requires more pronounced surface textures but can capture both the micro- and macro-surface shape using a fixed light source. It has been shown that combining SfM and Photometric stereo reconstructions can lead to better results in laboratory conditions [43].

Several approaches have been applied to evaluate the accuracy of SfM reconstructions. In [44], an approach from vision metrology was incorporated by using well-defined scan settings and reference targets to document the geometric accuracy. More often, approaches rely on a comparison to other optical techniques such as LiDAR, a laser scanner or a total station. In the case of reference point clouds as with LiDAR or a laser scanner, a direct comparison with the SfM point cloud through the calculation of signed or unsigned distances can be carried out [4,5,8,45]. Similarly, when digital elevation models (DEM) are produced, a direct raster-to-raster comparison is possible [22,45,46]. Conversely, a comparison can be made between a point cloud and reference target points obtained with, e.g., a total station [2,39]. In all cases, the comparison is limited by the measurement uncertainty in the reference model, e.g., of reference points or pre-aligned point-cloud [46]. A way to alleviate this is obtaining replicas of the surface by, e.g., replication molding and producing a highly accurate reference DEM using optical microscopy [23]. In addition, the use of 3D-printed objects allows for direct comparison of the measured geometries to the design geometry of the used CAD model [47].

To quantify the reconstruction accuracy, quantitative parameters or metrics are needed. Common metrics for evaluating SfM reconstructions are the standard deviation (SD) [1,2,5] or root mean squared (RMS) deviation [36,39,45] between SfM reconstructions and reference points. The SD and RMS metrics have usually been reported as overall global statistics. An advantage of the SD and RMS metrics is that negative and positive deviations will not cancel each other out as with the sometimes reported mean deviation or mean error [26]. As the accuracy might vary locally with the object surface geometry, metrics that capture spatial information across multiple length scales are of interest when evaluating SfM reconstructions. One way is by estimating an SD value for either each point on the point cloud or in a moving window around each point on the DEM [10,22,46,48,49].This can be used to study, e.g., the effect of local surface height—either absolute value or variation—on the reconstruction accuracy. Another way is power-spectral density (PSD) analysis, which gives a multi-scale quantification of the surface height variation [50,51]. Areal roughness parameters that describe different length scales were introduced by [23] for comparison between regions of interest in SfM and reference DEM. In addition, some studies have compared geometric measures on the object such as distances or angles [6,52] to the SfM point clouds. Notwithstanding, more systematic approaches for studying the accuracy dependence on local topography and surface roughness are needed.

**Contributions:** In this study, we present the first quantitative investigation of the influence of surface texture on SfM reconstructions. To isolate the effect of texture from other factors, the SfM measurements were performed indoors using two sets of reference artifacts as shown in Figure 1. These artifacts are easily replicable and use 3D-printed and easily obtainable materials. To study the effect of varying color contrast, a 3D-printed step-height artifact was prepared with different color patterns applied to the surface. The effect of local height variations was studied using a set of artifacts with industrial sandpaper of varying grit size covering the surface. The artifacts were chosen to provide quantitative reference values, i.e., the nominal height of the steps and the sandpaper grain size distributions are in accordance with ISO 6344 [53]. In addition, the use of a 3D-printed artifact allows for direct comparison to the design geometry. The influence of texture was compared to the effect of the camera-to-object range on the reconstruction.

We organize the surface analysis presented in the paper as a baseline that can be used for comparative studies of the performance of SfM. For optimal reconstruction quality, the SfM capture settings were designed to ensure a sufficiently small GSD of around 0.03 mm, which would be smaller or similar to the sandpaper grain sizes. To achieve this, SfM reconstructions were based on high-resolution images taken with a DSLR camera with a focal length of 260 mm and at a close range of around 2 m. In addition, the small GSD allows for comparing the SfM reconstructions to DEMs measured using calibrated confocal (CM) and focus variation microscopes (FVM).

To quantify the texture influence, we investigate a number of previously reported parameters and metrics that describe the reconstructed topography on both global, local, or multiple scales. These parameters include RMSE values, heights of surface structures, surface roughness values as well as PSD analysis.

We approach the investigations with the following hypotheses:For a reconstruction without erroneous areas, the GSD of the capture must be comparable to the spatial variation of the surface texture.Specifically for topographic or roughness texture, the GSD should be smaller than the roughness length scale.Multi-scale metrics are needed for a thorough analysis of the texture influence on the reconstruction.

Exploring these hypotheses can improve the understanding of the interplay between surface texture and reconstruction quality in SfM. Furthermore, the proposed metrics may provide quantitative guidelines for SfM settings when reconstructing low-texture regions or objects.

## 2. Methodology

In this section, we will introduce our step-by-step process for verifying the influence of surface textures and shapes on the quality of 3D reconstructions. We first discuss the two proposed custom artifacts used for measuring the micro- and macro-surface details and shapes. We then give an overview of the used capturing setup and SfM software solution. As we want to generate a highly detailed baseline, we use a high-resolution DSLR camera and SfM software which has proven to capture detailed representations of the imaged surfaces. Finally, we go into detail on the microscopy measurements used to create ground truth for both artifacts as well as the captured measurements and analysis tools such as step height overview, edge and particle resolution, roughness, and power spectral density analysis.

### 2.1. Preparation of Artifacts

Two types of samples were used: a 3D-printed step-height artifact with various coloring added and a set of artifacts consisting of foam pieces covered with sandpaper of grit sizes from P40 to P240. The former type represented an ordered height variation on the macro-level, while the latter represented randomly distributed height variations on the micro-level.

The step-height artifact of 50 mm in width and 95 mm in length consisted of five symmetric steps of nominal heights of 0.63 mm, 1.25 mm, 2.5 mm, 5 mm, and 10 mm and a nominal width of 5 mm. The artifact, seen in Figure 2a, was 3D printed using strong PLA material with subsequent polishing for additional smoothing of the surface. The removal of material by the polishing procedure is estimated to be of the order of 0.01 mm, which would not change the geometry significantly. In addition to the monochrome surface, two types of coloring were added to study the influence of color contrast on the SfM reconstructions. The types of coloring were: no coloring, a projected light pattern, and a marked pattern painted on the surface of the artifact.

For the sandpaper artifact, industrial sandpapers of varying grit sizes from P40 to P240 were attached to curved foam surfaces that were cut using a CnC foam cutter as shown in Figure 2b. The grit-size range was chosen to study the influence of surface roughness on the SfM reconstruction. As the sandpaper follows the standards proposed by [54], the grain size and hence the roughness are well-defined. Furthermore, sandpaper is readily available and can be easily added to different surface shapes and sizes. Finally, it has been shown that sandpaper grit can be used to approximate and describe the surface details and damages [55]. The average particle diameters of the used grit sizes are shown in Table 1.

### 2.2. SfM Capturing Conditions

An indoor environment was chosen for image acquisition for both types of artifacts. A Canon 5D DSLR camera with a Canon 70-300 f/4-5.6L IS USM zoom lens was used, and the captured images had a size of 8688 × 5792 pixels. The sensor pixel size was 4.14 μm and a focal length of 260 mm was used. The lighting was provided by two Elinchrom 4RX flashlights. The exposure time was set to 1/200 s so problems of blurring caused by vibrations were prevented and the F-stop was f/20, so all the surface of the imaged artifacts was in focus.

The range, i.e., the sample-to-camera distance, was 1.5 m for the three texture levels of the step-height artifact and 1.7 m for the seven grit sizes of the sandpaper artifact. To investigate the influence of distance, the range was varied for a subset of the artifacts: at 1.5 m and 2.0 m for the step-height artifact colored with a marked pattern, and at 1.5 m, 1.7 m, 2.0 m, and 2.2 m for the P40 sandpaper artifact. Here, 1.5 m was chosen as a lower limit on the range since the chosen lens had difficulties focusing at closer distances. The upper limit was restricted to 2.2 m by the size of the indoor location. The range settings are summarized in Table 2.

If the focal length has been fixed, the size of the object that each pixel in the image represents can be calculated using the ground sampling distance (GSD). To calculate the GSD, we can use the already known pixel size, the range to the surface, and the focal length. The calculated GSD values for ranges between 1.5 and 2.2 m are shown in Table 2.

To capture all surface details of the two types of artifacts, images were acquired at every 10 degrees along a semi-circular path at 180 degrees around the artifact. This was repeated for three different heights with the camera facing towards the artifact at all times resulting in a total of 54 images. As the same number of images was used regardless of the range, the overlap of individual images will vary in these reconstructions. An example of the capturing paths around the sandpaper blade artifact can be seen in Figure 3b.

### 2.3. From Point Cloud to Topographic Map

As we want to introduce a baseline that would represent the best possible case for SfM, we choose to go with the ContextCapture SfM reconstruction software by Bentley [56]. It provides very high accuracy while still being sensitive to sub-optimal capturing and object surface conditions [32]. It was chosen instead of other open-source alternatives, as they provide a more mixed performance, depending on the capturing environment, which would introduce noise and uncertainties in the proposed analysis pipeline. This way, the influence of the artifact textures and surfaces on SfM can be better captured. Figure 3 represents the SfM reconstruction pipeline as well as extracting and rasterizing patches. Initially, all the images for a single artifact were imported in ContextCapture Figure 3a. The 2D features from these images were then extracted, matched, and the ones deemed noise were discarded. From these feature matches, the 3D positions of points in a sparse point cloud, together with initial camera positions were calculated, as seen in Figure 3b. Next, the sparse points and camera positions were used to calculate depth maps for each image, which in turn were utilized to densify the point cloud and bring back additional details and calculate colors for each 3D point, as seen in Figure 3c,d. The resultant point clouds have arbitrary scales, so to establish the absolute real-life scale, the reference lengths on the artifacts were used. For the foam artifacts, the scale was fixed using the distance between a set of feature points on the surface. For the step-height artifact, the length and width of the artifact were used to fix it to an absolute scale.

To go from point clouds to a more easily analyzable structure, a digital elevation model (DEM) was created from selected patches of each of the artifacts using CloudCompare [57]. The reconstructed sandpaper samples were registered to each other so a patch can be extracted from each from approximately the same place. The registration was completed by minimizing the distance between them using an iterative closest point (ICP) algorithm, and a patch was extracted from each reconstruction Figure 3e. For the sandpaper artifacts, the patches were roughly 50 mm × 50 mm in size, while the full 50 mm × 95 mm area of the step-height artifact was extracted. All patches were oriented in a way that their Z-axis was pointed in an up-direction and rasterized by projecting the mean Z height value for each point to a grid surface, as seen in Figure 3f.

### 2.4. Microscopy Measurements

For the microscopy reference measurements of the step-height artifact, the 3D surface geometry was acquired through focus-variation microscopy (FVM) using a calibrated Hirox RH-2000 microscope [58]. A region-of-interest topographic map was measured across each step. The 0.63 mm, 1.25 mm, and 2.5 mm steps were measured using the MXB-5000REZ mid-range objective with ×140 magnification, a pixel size of 2.18 µm, and a 22.5 µm vertical step size. The MXB-2016Z objective with ×100 magnification, a pixel size of 1.69 µm, and a 150 µm vertical step size were used for the 5 mm and 10 mm steps. For the 3D surface reconstruction, the custom Hirox software was used. The FVM microscope was calibrated in the vertical direction to a set of gauge blocks with traceability through a laser interferometry calibration. The relative uncertainty was 2% in the vertical direction.

As reference measurements for the sandpaper artifacts, confocal microscopy (CM) measurements were carried out on sandpaper samples of the sandpaper used for the foam artifacts. A calibrated PLU NEOX confocal microscope by Sensofar [59] with ×5 magnification was used. The spatial pixel size was 3.32 µm, and the vertical step size was 12 µm. The reconstruction from the microscopy images was achieved through the SensoSCAN software [60]. Two sets of measurements were conducted: one with a square FoV of around 10 mm × 10 mm and one with a rectangular FoV of 40 mm × 2 mm. The former was used for particle size analysis and the latter for power spectral density analysis. The CM microscope was calibrated using step height transfer standards for its height capturing, and the achieved relative uncertainty was 3% in the vertical direction.

## 3. Data Analysis

For analyzing the captured microscopy reference measurements and the comparative SfM ones, both MATLAB and the Scanning Probe Image Processor (SPIP) [61] were used. SPIP is a widely used software solution for microscopy image analysis because of the built-in image structure tools. Each of the calculated comparative properties from the microscopy and SfM reconstructions is explained below.

### 3.1. Height of Steps

The step height was calculated from the DEMs using the “ISO 5436 Step Height” [62] analysis tool of the SPIP application software. In turn, each step of the artifact was selected and cropped with the step aligned along the up–down direction. A leveling was made using ground level on both sides of the step, and an “ISO 5436 Step Height” analysis was performed on each line across the step. The result was a mean value as well as an SD for the height of each step.

### 3.2. Edge Resolution

As a measure of the spatial resolution of the step-height artifact, the edge resolution (ER) was used. Each step of the artifact was divided into sections. For each section, the ER was found as the width between the 10% and 90% height level. Thus, the ER is zero for a 90-degree side wall angle and grows when the angle decreases. The mean value and standard deviation of the ER were calculated for each step.

### 3.3. Particle Analysis

Ideally, the individual grains of the sandpaper samples would appear in the SfM and CM reconstructions as separable particles. Thus, by analyzing the particle size, a comparison to the nominal average grain size should be possible. Therefore, to evaluate the SfM reconstruction of the sandpaper, image segmentation was carried out on the SfM DEMs and CM topographic maps to label individual particles.

The initial image segmentation was conducted in four steps. First, the topographic map was filtered to remove low-frequency components. A Gaussian filter with a 5 mm cut-off wavelength was applied using the SPIP software. Secondly, an adaptive thresholding segmentation of height structures and background was performed using the MATLAB image processing toolbox. For each pixel, the segmentation finds a threshold level from a local neighborhood analysis. Thirdly, a modified watershed algorithm for irregular features from the Biovoxxel toolbox [63] in ImageJ was applied to label individual particles in a label map. For the algorithm, a convexity level of 0.95 was used. Lastly, using the label map, each individual particle could be selected from the topographic map. In Figure 4, the segmentation process is illustrated.

From the image segmentation, the height of each labeled particle was calculated using MATLAB. The height was found as the distance from the highest point on the individual particle to the background reference plane. The background reference plane was found as the global mean height level of the background in the topographic map. A particle density value was calculated as the ratio of the number of labeled particles to the measured surface area.

### 3.4. Roughness Analysis

The root mean square area roughness parameter $S_{q}$ was chosen as a measure of the overall height variations of a DEM. The $S_{q}$ parameter of the DEM was calculated using the SPIP’s “Roughness Analysis” tool. At first, the surface was leveled. Then, an area roughness analysis was carried out using a S-filter of 10 μm and an L-filter of 5 mm as described in [64].

### 3.5. Power Spectral Density Analysis

To investigate the wavelength dependence of the SfM reconstructions, a power spectral density (PSD) analysis was conducted on the sandpaper-artifacts measurements. When used on measurements of surface topography, the PSD is the Fourier transform of the surface height autocorrelation function. More in-depth introductions to the use and calculation of PSD of surface topography measurements can be found in [65,66]. A PSD analysis was carried out on both CM and SfM measurements of the sandpaper artifacts and the SfM topographic maps of both step-height and sandpaper artifacts for the different capturing conditions, i.e., object texture and imaging range. As an output of the PSD analysis, an average of one-sided 1D PSD curves from the lines in the topographic maps was calculated. The software used was the “Average X-PSD” tool of the "FFT/PSD analysis" toolbox in SPIP.

## 4. Results

### 4.1. Step-Height Artifact

The variation in color contrast on the step-height artifact had a clear impact on the SfM reconstructions. For the smooth monochrome surface, the measurement at 1.5 m resulted in not enough feature points being extracted, and the ContextCapture SfM software could not produce neither a point cloud nor a mesh. This was improved somewhat by projecting a random dot pattern perpendicularly on the surface using a light projector. As seen in Figure 5c, a reconstruction of the artifact was possible, although it suffered from a few holes and variations in the side-wall widths. An even better improvement was achieved when the horizontal and vertical faces of the surface were marked with a pattern of spots and lines. Figure 5a,b show the reconstructions of the marked pattern condition at a range of 1.5 m and 2 m, respectively.

Qualitatively, the influence of the color contrast is visible in the middle and bottom rows of Figure 5. A much larger variation in width is observed with the projected light pattern of Figure 5c (middle panel) than for the marked pattern in Figure 5a. In addition, while the line profile of the marked pattern reconstruction in Figure 5a (bottom panel) shows almost rectangular side walls and step corners, the side wall and corner angles are reduced in the line profile of the projected pattern in Figure 5c. This indicates a loss in the resolution and accuracy of the reconstruction. A somewhat similar effect is seen by increasing the range from 1.5 to 2 m as seen by comparing Figure 5a,b.

Quantitative results are shown in Table 3 and Table 4. Since no successful SfM reconstruction was obtained from the monochrome smooth surface, too little color contrast clearly affects the measurement of height values. However, for all successful SfM reconstructions and FVM measurements, the mean height values were comparable regardless of the variation in color contrast and imaging range, as shown in Table 3. In contrast, the capturing conditions did influence the SD of the measured height values, which is further illustrated in Figure 6A. The use of a projected light color contrast at a 1.5 m range gave a significantly larger SD compared to the marked pattern measurement. On average for all steps, an increase of a factor of 5 was observed. In addition, moving the marked pattern artifact to a range at 2 m also increased the SD values by a factor of 1.5 on average. The RMSE value between the full SfM reconstruction and design geometry of the artifact showed a similar behavior, as seen from Table 4. The RMSE was 1.8 times larger for the projected pattern and 1.2 times larger for the marked pattern at 2 m compared to the marked pattern at 1.5 m range. While in general, the SD values for the smaller steps were relatively constant within each measurement, a significant increase was seen especially for the 10 mm step. For the projected light pattern, the 10 mm step SD value had to be cropped at 0.5 mm in Figure 6A to allow for a visual comparison of the other results.

The lateral spatial resolution was also influenced by the capturing condition as illustrated by the edge resolution (ER) in Figure 6B. As recalled from Section 3.2, the ER is the lateral width between the 10% and 90% height of the step. The lowest ER values were found to be between 0.4 and 0.5 mm. These were for the 5 mm and 10 mm steps with a marked pattern at 1.5 m. Both the use of the projected light pattern and increasing the range with the marked pattern from 1.5 to 2 m resulted in a reduced resolution as seen from the larger ER. In both cases, the ER across all steps was on average a factor of two larger compared to the 1.5 m marked pattern measurement. While the 10 mm step had the largest SD height value, the ER values were largest for the 0.63 mm step and in general decreased with the height. Thus, both low and high aspect ratios of the steps can pose a challenge for the accuracy of the reconstructed geometries by SfM.

How the capturing conditions affected the reconstructions can be seen in more detail from the PSD analysis in Figure 6C. The dotted (blue) line shows a simulated PSD curve using the nominal artifact geometry. The full (red), dashed (yellow) and dot-dashed (purple) lines are from the SfM measurements using a projected pattern at 1.5 m, which was marked at 1.5 m and 2 m, respectively. For the long-wavelength components at frequencies from 0.01 to 0.1 mm${}^{-1}$, the SfM curves follow the nominal PSD. Thus, all reconstructions are able to reconstruct the large-scale geometry of the step-height artifact.

However, from 0.3 mm${}^{-1}$ and depending on the capturing conditions, the SfM curves start to deviate from the nominal. The marked pattern reconstruction at 1.5 m follows the closest path to the nominal curve. The distinct peaks at high frequencies in the nominal curve can also be found in the SfM curve up to around 3 mm${}^{-1}$. Increasing the SfM range to 2 m is seen to cause a reduction in the PSD intensity from 0.3 mm${}^{-1}$ and above compared to the 1.5 m curve. In addition, the distinct peaks are only visible up to 2 mm${}^{-1}$. This indicates that an increased range decreases the sensitivity of the reconstruction across a range of frequencies and reduces the resolution limit.

For the projected light pattern reconstruction, a different behavior is seen. From 0.3 mm${}^{-1}$, the peak-to-valley amplitude of the curve oscillations decreases rapidly, and from around 1.2 mm${}^{-1}$, the curve flattens. Furthermore, the intensity level lies above the marked pattern curve from 1 mm${}^{-1}$ and even the nominal curve from 4 mm${}^{-1}$. This larger intensity could indicate an increased noise level. According to [66], the presence of white noise introduces a flat region at high frequencies of a PSD curve with an intensity proportional to the strength of the noise. Thus, an interpretation could be that the degradation in color contrast leads to increased noise in the reconstruction, which was not seen when changing the range.

### 4.2. Sandpaper Artifacts

The influence of the topography texture on the SfM reconstructions could be seen in both the reconstructed color and geometry. In Figure 7, the color contrast of the SfM reconstructions from P40 to P240 at a 1.7 m range are shown. While the color contrast of P40, P60, P80, and P100 reconstructions appears uniform across the patches, darker areas are visible on the P120, P180, and P240. These artifacts could be caused by too few detected feature points due to the reduced topography texture.

The reconstructed SfM topography can be studied more closely in Figure 8. Figure 8D through Figure 8F show a zoom-in of the SfM DEM of grit sizes P40, P80, and P180, respectively. The corresponding CM topographies are shown in Figure 8A through Figure 8C. The image width has been scaled down with a factor of two between each column to mimic the decreasing grain diameter as shown in Table 1.

As seen from the figure, the SfM reconstructions suffer from a much poorer spatial resolution compared to the CM measurements. While the lateral resolution of the CM measurement was limited by the pixel spacing of 0.03 mm, the SfM resolution from the ER analysis in Section 4.1 was found at best to be 0.5 mm. Accordingly, a direct comparison of SfM and CM would be difficult. To compare the two topographies with very different resolutions, an alternative approach inspired by [67,68] was employed. By applying a Gaussian low-pass filter to the CM topographies of Figure 8A through Figure 8C, a set of topographic data with spatial resolution resembling the SfM DEM can be modeled. The filter width was determined by applying a range of Gaussian filters to synthetic data of an ideal edge to find the relation between the values of the ER and σ. An ER value of 0.5 mm was chosen which gave σ = 0.21 mm. The resulting topographies after applying the Gaussian filter are shown in Figure 8G through Figure 8I. Especially for the P40 and P80 grits, the filtered CM (fCM) topographies have a similar appearance and height range as the SfM topographies. For the P180 grit, however, the SfM DEM of Figure 8F shows a larger height range as well as a higher level of high-frequency noise than the fCM of Figure 8I.

For the particle analysis, the limited spatial resolution of the SfM DEMs meant that the grains could not be resolved individually. Instead, groups of neighboring particles were observed as illustrated by comparing fCM topographies in Figure 8G through Figure 8I to CM in Figure 8A through Figure 8C. In addition, the resolution was also found to affect the measured heights. While the CM particle size distributions in Figure 9B range from 0 to 600 μm, both the SfM of Figure 9A and fCM distributions of Figure 9C lie in the range from 0 to 400 μm. Nonetheless, the particle analysis demonstrated that the fCM data also quantitatively gave a better description of the SfM topographies. Figure 9D shows that the mean particle heights are also fairly comparable for fCM and SfM albeit with some discrepancies for grits P40, P180 and P240. Since the CM values were in good correspondence with the nominal grain sizes, this supports that the smaller height values found by the SfM were due to the limited lateral resolution.

Table 5 summarizes the mean height and particle density from the particle analysis as well as the $S_{q}$ parameter from the area roughness analysis. The particle density values for SfM reflect how the reduced lateral resolution limits the detection of individual grains. While the CM density is seen to be around 2.5 times larger than the SfM value for P40, this difference increases to a factor of around 50 for the P240 grit. The RMS roughness $S_{q}$ provides an alternative measure of the overall height variations compared to the particle analysis. Nonetheless, the $S_{q}$ values from both CM, SfM and fCM data followed the same trend as the average particle height. In fact, the correlation coefficient between the mean height from the particle analysis and $S_{q}$ was 0.99. As with the mean height, the $S_{q}$ values for SfM and fCM were comparable despite some discrepancies for grits P40, P180 and P240.

The PSD analysis of Figure 10 offers a look at the frequency dependency of the CM, SfM and fCM topographies. The figure shows the PSD intensity in the frequency interval 0.1 to 10 mm${}^{-1}$, which corresponds to wavelengths from 10 mm and down to 0.1 mm, respectively. For the CM curves, a flat region is seen at low frequencies in the interval 0.1 mm${}^{-1}$ to 1.0 mm${}^{-1}$, which is a common feature of PSD curves of engineered surfaces [65,66]. The frequency where the curve begins to decrease, called the roll-off wave vector, is related to the largest structures of the surface [65,66]. As seen by comparing Figure 10A through Figure 10F, the roll-off of the CM curves moves to higher frequencies which correspond to the smaller grain size of the finer grits.

The same feature is not observed in the fCM curves. The reduced resolution due to filtering is seen to lower the PSD intensity of the fCM curves and especially at the higher frequencies. Consequently, no roll-off wave vector is observed for the fCM curves. Similar behavior is seen for the SfM curves which overlap with the fCM curves for P60 to P100, as seen from Figure 10A through Figure 10C. However, at P120 and finer grit sizes, the PSD intensities of the SfM reconstructions increase above the fCM intensities, as seen in Figure 10D through Figure 10F. This increase reflects the presence of topography variations in the SfM reconstructions that cannot be fully explained by the model of reduced resolution. One explanation could be that the texture level becomes too low at P120 for the SfM reconstructions to function properly. As an effect, an increased amount of noise appears in the reconstructed topography, which was also observed in Figure 8F. The presence of increased noise could also explain the SfM discrepancies in particle height and $S_{q}$ for the P180 and P240 grit sizes.

For the P40 sandpaper artifact, SfM measurements at four different distances were conducted at ranges of 1.5 m, 1.7 m, 2.0 m, and 2.2 m. As the range is increased, a proportionally larger spatial resolution should be observed. To include this effect in the modeled fCM data, the P40 CM measurement was filtered using Gaussian low-pass filters of increasing filter size. Thus, a value of σ proportional to the range was applied giving a σ of 0.19 mm, 0.21 mm, 0.25 mm and 0.27 mm.

The quantitative parameters from the range study are summarized in Table 6. With increasing range, a smaller average height, particle density, and $S_{q}$ roughness were observed for the SfM data. A similar decrease of parameter values was seen with an increasing filter size of the fCM data. As was also observed for the 1.7 m data in Table 5, the SfM height and $S_{q}$ values of Table 6 are seen to be smaller than the fCM values for all ranges. These differences indicate that the modeled fCM data does not give a full description of the SfM reconstruction of the P40 grit.

Figure 11 shows the height distributions from the particle analysis of the P40 grit. In Figure 11A, the SfM height distributions at varying range are shown, while Figure 11B shows the height distributions of the modeled fCM data. With increasing range, the height distributions are seen to shift toward lower height values. A similar behavior can also be seen for the increasing filter width. However, although the same trend is observed, the fCM mean height values are larger than the SfM values as shown in Figure 11C.

The spatial frequency information of the PSD analysis also shows a dependence on the range. As seen from Figure 12, an increase in range causes a reduction of the PSD intensity of the SfM curves in the frequency interval 0.5 to 2.0 mm${}^{-1}$. As an example, the 1.5 m range SfM curve has an intensity close to $10^{-3}\;{\mathrm{mm}}^{3}$ at a frequency of $1.0\;{\mathrm{mm}}^{-1}$, while the intensity for the 2.2 m curve has decreased to just above $10^{-2}\;{\mathrm{mm}}^{3}$. The changing filter size of the fCM curves is seen to result in similar behavior. Nonetheless, the overlap between fCM and SfM is not seen to be perfect. Meanwhile, the PSD intensity for the fCM curves is smaller between 1.0 and 2.0 mm${}^{-1}$, and a larger intensity is observed for frequencies between 0.2 and 1.0 mm${}^{-1}$.

## 5. Discussion

Overall, the results show that some level of texture is required in SfM reconstructions. This was illustrated by the scan of the monochrome step-height artifact where the reconstruction failed altogether. In addition, this study demonstrates that the surface texture also heavily influences the reconstructed surface topography of SfM DEMs. In comparison to the marked pattern step-height, the low-level color-contrast of the projected light pattern resulted in a lower precision of the measured height as well as a reduced lateral resolution. However, whereas [44] found that a projected light pattern decreased the accuracy compared to no pattern, the projected light in this study did aid in the SfM reconstruction. At low roughness-levels of the sandpaper artifact, reconstruction errors in the color texture were observed.

Furthermore, the PSD analysis of both artifact types indicated an increase in noise at low texture levels. This was observed as a larger PSD intensity at high spatial frequencies for the projected light pattern of the step-height artifact and for finer grit sizes of the sandpaper artifact. In this way, the PSD analysis indicated when the texture level became a significant source of high-frequency noise. In the case of the sandpaper artifacts, a transition was seen between P100 and P120. Thus, the roughness texture became insufficient for a nominal average grain size between 125 and 160 μm corresponding to five to six times the GSD of 27 μm.

The spatial resolution in the SfM scans was found to be around 0.5 mm. This was seen from the best-case ER values from the marked step height as well as the fCM model used for the sandpaper data. The best-case ER values are used, as these give the best estimate for the achievable resolution without other factors influencing the ER. Interestingly, the 0.5 mm resolution means that the texture transition level was below the lateral spatial resolution. This indicates that the texture is important for the reconstruction even though the variations themselves cannot be spatially resolved. As observed here, the spatial resolution was more than an order of magnitude larger than the GSD, which has also been reported by other high-resolution SfM studies [23,51]. When the size of surface structures was close to the spatial resolution, the error in the reconstruction increased as observed from the ER in the step-height artifact. Below the resolution, individual sandpaper grains could not be accurately resolved.

The importance of texture level poses a challenge when evaluating the quality of SfM reconstructions. Commercial software may produce a reconstruction even at low levels of texture. As seen in this study, these reconstructions could introduce erroneous texture features and increased geometrical variations. Although such deviations may be possible to detect when measuring known surfaces, the effect of low texture could lead to false interpretations when inspecting unknown surfaces. Thus, the level of texture on the object surface should be taken into account when applying SfM for geometrical measurements.

The influence of spatial resolution of the SfM reconstructions was probed by a variation in range. Keeping a constant focal length, changing the range from 1.5 to 2.2 m increased the GSD from 24 to 35 μm. At a larger range, the SfM reconstructions of the sandpaper artifact resulted in smaller particle height and $S_{q}$ values. For the marked pattern step-height artifact, an increase in range from 1.5 to 2.0 m resulted in less precise reconstructions as seen from the larger height SD, RMSE, and ER values. The RMSE values of 0.57 mm at 1.5 m and 0.70 mm at 2.0 mm are comparable to previously reported values [36]. However, the low texture of the projected light pattern led to a doubling in RMSE of 1.04 mm at 1.5 m range. For the PSD analysis, an increase in range led to smaller PSD intensities at high spatial frequencies for both artifact types. This was in contrast to the observations at low texture levels where increased intensities were observed. Thus, the PSD analysis might indicate whether a SfM measurement is limited by insufficient spatial resolution or surface texture.

## 6. Conclusions

With this study, we demonstrate how the accuracy of 3D reconstructions using SfM changes is based on the surface texture of the imaged objects. In SfM, while attention is often given to the environment, lighting, and camera setup, the requirements for the imaged surfaces are normally only inspected for transparency and reflectivity. We demonstrate through an in-depth comparative analysis between microscopy and SfM reconstructions that both micro- and macro-surface texture and shape details have a strong influence on the quality of the produced 3D models. This was inspected using a range of quantitative metrics. For micro-surface textures, we show that the SfM reconstructions begin to struggle when the spatial size of the surface roughness becomes smaller and approaches the GSD. Erroneous areas begin to appear in the reconstruction, and the overall noise level increases. We observe a transition at around five to six times the GSD, which could be used as a guideline when setting up measurements using SfM. Furthermore, PSD analysis could be a way to probe for sufficient texture by inspecting the high-frequency end for noise components.

A range of local, global, and multi-scale quantitative metrics were investigated. While an effect of low texture could be observed in both global RMSE or more local Sq, particle size, ER or height values, the most insight was gained from the multi-scale PSD analysis. This made it possible to distinguish behavior at both large and small spatial scales.

The observed spatial resolution was 10–20 times larger than the GSD, which illustrates that a sort of low-pass smoothing takes place as part of the SfM reconstruction. Furthermore, the spatial resolution was larger than the observed transition at five to six times the GSD, which indicates that sub-resolution features are important for the overall reconstruction.

As a future study, the influence of color texture could be investigated more thoroughly using a range of color patterns of varying areal coverage, size, and shape. This might determine whether a transition level of insufficient color contrast exists. The type of patterns should preferably be marked directly on the surface rather than projected using a light projector. In addition, other state-of-the-art SfM software solutions could be investigated to determine whether the influence of texture depends on the software or is a fundamental property of SfM.

## Figures and Tables

**Figure 1 sensors-23-00178-f001:**
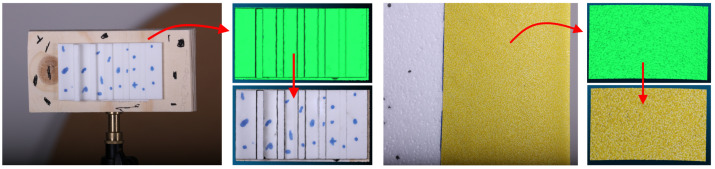
Step-height and sandpaper artifacts used in the paper as well as the reconstructed meshes and color textures.

**Figure 2 sensors-23-00178-f002:**
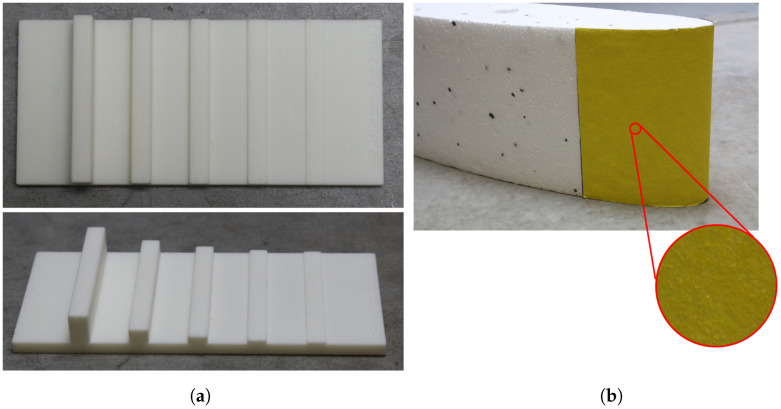
Artifacts for validating reconstructions of the surface topography from macro- and micro-height variations. (**a**) shows the 3D-printed step artifact with nominal heights: 0.63 mm, 1.25 mm, 2.5 mm, 5 mm, and 10 mm. (**b**) shows an example of a foam artifact with the P40 sandpaper attached to it.

**Figure 3 sensors-23-00178-f003:**
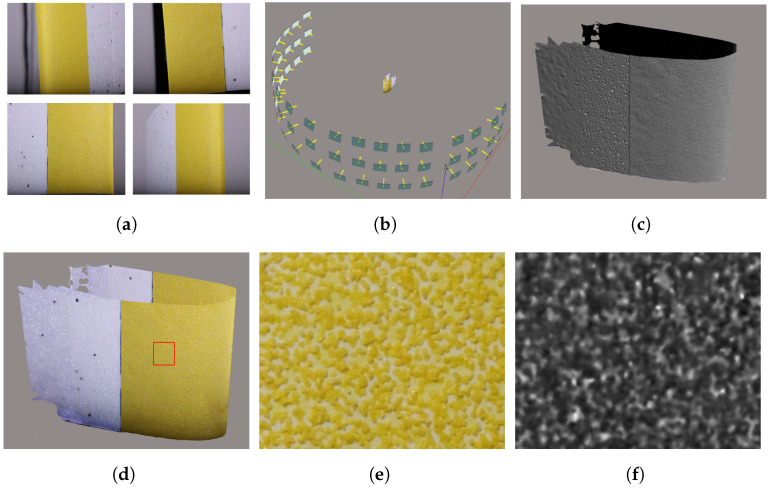
The described photogrammetry pipeline. As an input, ContextCapture takes the images (**a**), which were captured at 10-degree intervals in a semi-circle around each artifact in three heights. Features are extracted from each image, matched and a sparse point cloud and camera locations are computed; see (**b**). These are then densified and the RGB colors for each point cloud are calculated, as shown in (**c**,**d**). Finally, patches are extracted from each artifact and rasterized into depth maps; see (**e**,**f**). The subfigures are as follows: (**a**) Initial images captured from different positions. (**b**) Calculated camera positions. (**c**) Result mesh without texture. (**d**) Result mesh with texture. (**e**) Extracted patch close-up. (**f**) Rasterized depth map.

**Figure 4 sensors-23-00178-f004:**

Illustration of the image segmentation of P80 sandpaper. (**A**) The topographic map after filtering to remove low-frequency variations. (**B**) The segmentation result after the adaptive thresholding is shown in green overlaid on the topographic map. (**C**) The coloring shows the labeling of the individual particles after applying the modified watershed algorithm to the segmentation. The width of the scale bar is 0.5 mm.

**Figure 5 sensors-23-00178-f005:**
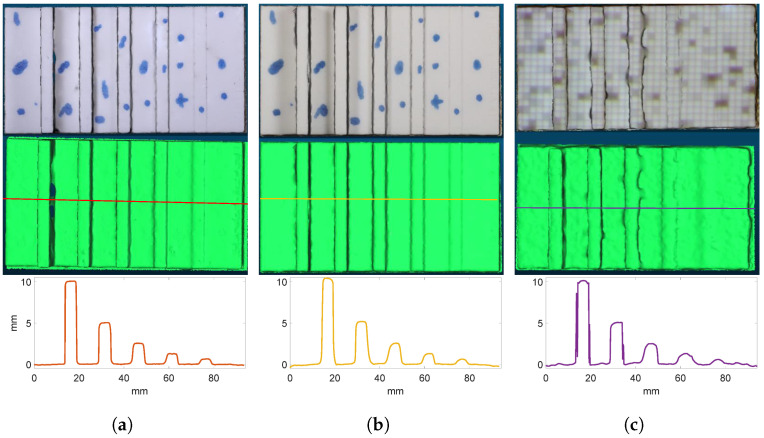
The reconstructed step-height artifact. The reconstructions in (**a**) at 1.5 m and in (**b**) at 2.0 m range are of the manually marked artifact. In (**c**), the reconstruction using a projected pattern over the artifact is shown. The top and middle row shows the reconstruction with and without color texture. In the bottom row, a line profile is shown across the artifact with the position indicated by the line in the middle row.

**Figure 6 sensors-23-00178-f006:**
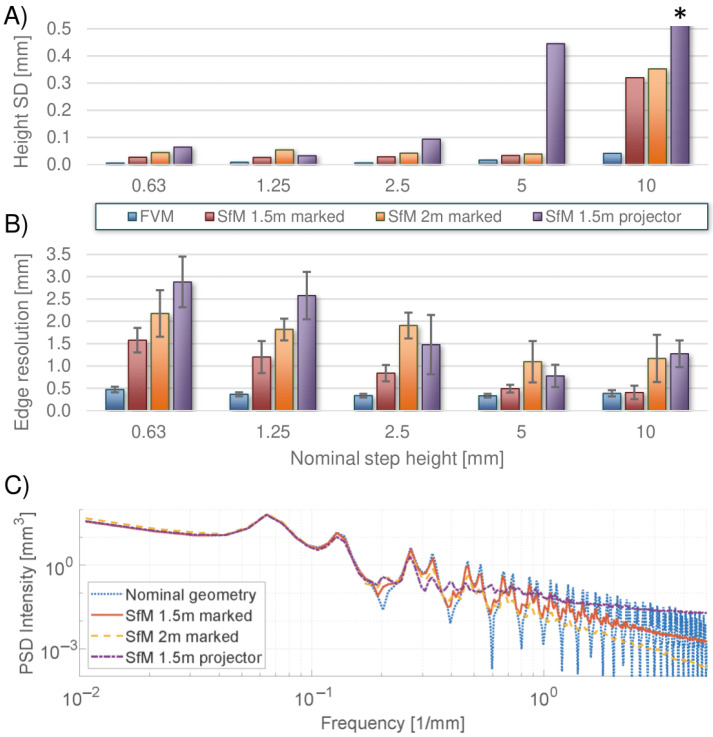
Step height artifact parameters from FVM, SfM marked pattern at 1.5 m and 2.0 and SfM projected pattern at 1.5 m. (**A**) SD of measured step height. Note that the SD of the 10 mm step of the SfM projected pattern was cropped off at 0.5 mm for better visual comparison. (**B**) measured ER with the SD as errorbar. (**C**) Power spectral density (PSD) analysis of the full reconstructed step-height artifact. Both intensity and frequency are shown on a log-scale. SfM measurements are compared to a nominal PSD based on the design geometry.

**Figure 7 sensors-23-00178-f007:**
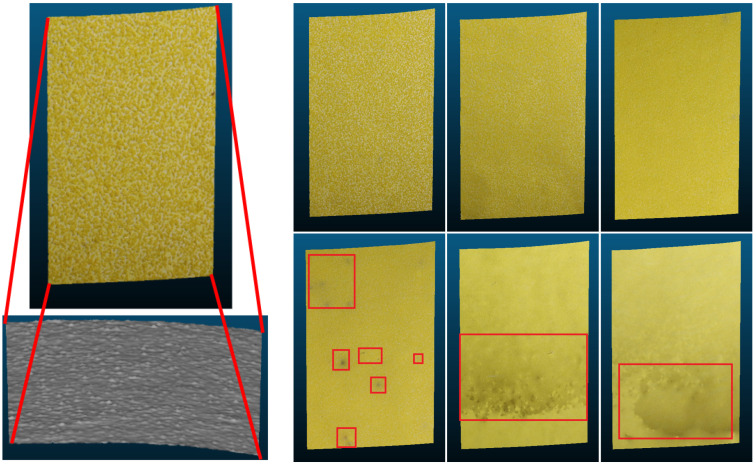
SfM reconstructions of the seven grit sizes at 1.7 m range. (Leftmost) P40 with close up of the geometry. (**Top** row) P60, P80, P100 and (**bottom** row) P120, P180, P240. The P120, P180 and P240 reconstructions show darker areas with erroneous color texture. These areas are marked with red.

**Figure 8 sensors-23-00178-f008:**
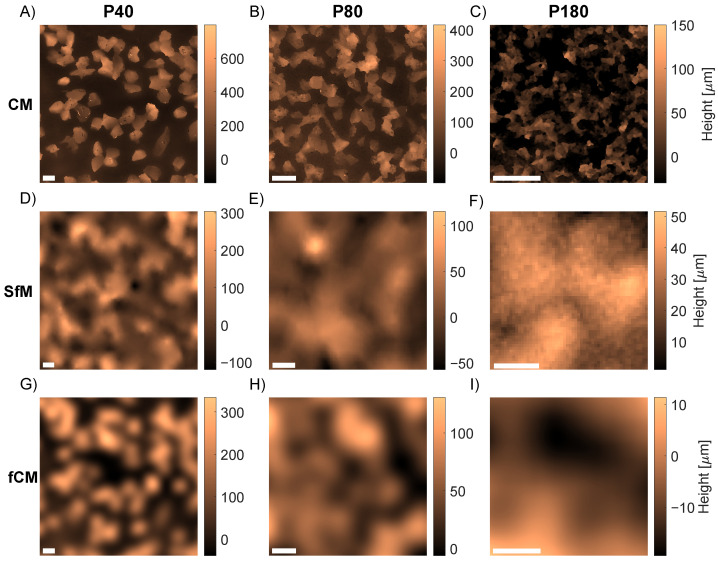
Topographic maps of sandpaper artifacts with grits P40, P80 and P180. CM (top row), SfM (middle row) at 1.7 m range, and fCM (bottom row) after Gaussian filtering with σ = 0.21 mm. (**A**,**D**,**G**) P40. (**B**,**E**,**H**) P80. (**C**,**F**,**I**) P180. The width of the scale bar is 0.5 mm in all panels. Note that CM and SfM have been measured at different sample locations.

**Figure 9 sensors-23-00178-f009:**
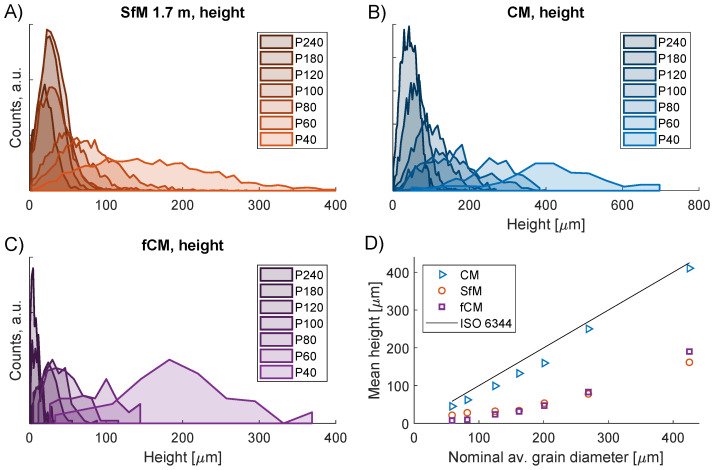
(**A**–**C**) Height distributions for SfM, CM and fCM particle analysis. (**A**) SfM, (**B**) CM, (**C**) fCM, (**D**) Mean measured height values versus the nominal grain diameter. The SfM range was 1.7 m.

**Figure 10 sensors-23-00178-f010:**
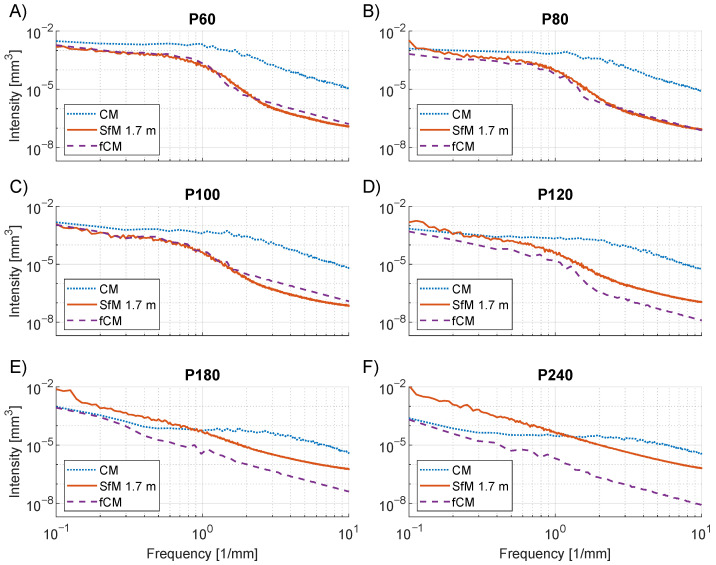
PSD analysis of CM (dotted line), SfM at 1.7 m range (full line), and fCM (dashed line) topographies of sandpaper artifacts of grit sizes P60–P240. The average of 1D PSD curves for each line in the DEM is shown on a log–log scale. (**A**) P60. (**B**) P80. (**C**) P100. (**D**) P120. (**E**) P180. (**F**) P240.

**Figure 11 sensors-23-00178-f011:**
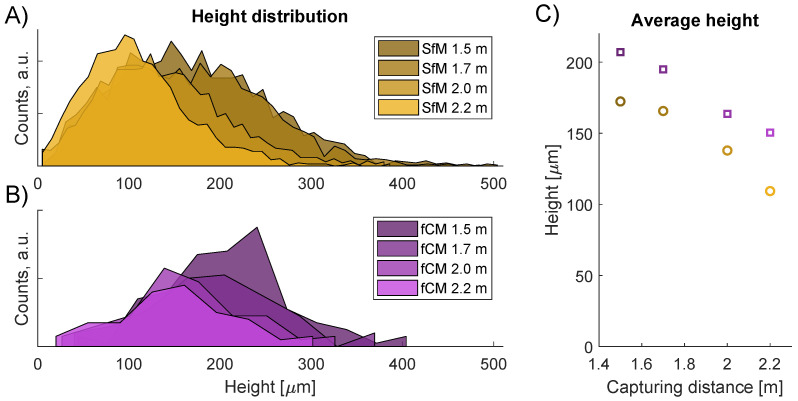
Particle height for P40 artifact at varying capture distance. (**A**) SfM distributions. (**B**) fCM distributions. (**C**) Mean height vs range.

**Figure 12 sensors-23-00178-f012:**
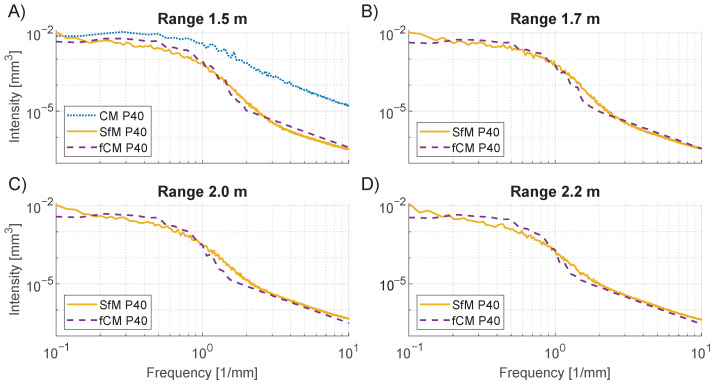
PSD analysis of P40 grit size at varying range or filtering level. CM (dotted line), SfM (full line), and fCM (dashed line). The average of 1D PSD curves for each line in the DEM is shown on a log–log scale. (**A**) 1.5 m. (**B**) 1.7 m. (**C**) 2.0 m. (**D**) 2.2 m.

**Table 1 sensors-23-00178-t001:** Sandpaper grit size and the nominal average particle diameter in μm [54].

Grit Size	P40	P60	P80	P100	P120	P180	P240
Nom. av. diam. (μm)	425	269	201	162	125	82	58.5

**Table 2 sensors-23-00178-t002:** Summary of the range and focal length used for sample scans and the corresponding GSD at the sample. P## = sandpaper artifact of grit P##, SH = step-height artifact, c = clean, p = projected light pattern, m = marked pattern.

Scans	SH_c_, SH_p_, SH_m_, P40	P40, …,P240	SH_m_, P40	P40
Focal length	260 mm	260 mm	260 mm	260 mm
Range	1.5 m	1.7 m	2.0 m	2.2 m
GSD	24 μm	27 μm	32 μm	35 μm

**Table 3 sensors-23-00178-t003:** Parameters from the step-height artifact measurements using FVM and SfM. SfM was conducted for marked pattern artifact at 1.5 m (SM1.5) and 2.0 m (SM2) and for artifact with projected light patterns at 1.5 m (SP1.5). Listed are mean height and ER values for each step. The standard deviations are shown in parentheses.

Step	Height				Edge Resolution (ER)			
	FVM	SM1.5	SM2	SP1.5	FVM	SM1.5	SM2	SP1.5
[mm]	[mm]	[mm]	[mm]	[mm]	[mm]	[mm]	[mm]	[mm]
0.63	0.620	0.629	0.588	0.61	0.472	1.58	2.18	2.88
	(0.0055)	(0.027)	(0.044)	(0.064)	(0.062)	(0.27)	(0.52)	(0.57)
1.25	1.225	1.253	1.21	1.183	0.365	1.12	1.82	2.58
	(0.0087)	(0.026)	(0.053)	(0.032)	(0.047)	(0.36)	(0.24)	(0.53)
2.5	2.468	2.519	2.523	2.45	0.335	0.84	1.91	1.48
	(0.0060)	(0.028)	(0.042)	(0.094)	(0.043)	(0.18)	(0.29)	(0.66)
5	4.959	5.042	5.095	4.81	0.334	0.492	1.10	0.78
	(0.016)	(0.033)	(0.039)	(0.45)	(0.042)	(0.087)	(0.46)	(0.25)
10	9.887	9.99	10.32	9.2	0.387	0.41	1.17	1.27
	(0.041)	(0.32)	(0.35)	(1.5)	(0.071)	(0.15)	(0.53)	(0.30)

**Table 4 sensors-23-00178-t004:** Raster-to-raster RMSE between SfM and design geometry. Marked pattern artifact at 1.5 m (SM1.5) and 2.0 m (SM2) and projected light pattern artifact at 1.5 m (SP1.5).

Scan	SM1.5	SM2	SP1.5
RMSE	0.57 mm	0.70 mm	1.04 mm

**Table 5 sensors-23-00178-t005:** Parameters from CM, SfM at a range of 1.7 m and filtered CM (fCM). Values shown are the average particle height, the particle surface density and the $S_{q}$ roughness parameter.

Grit Size	Av. Height			Density			$S_{q}$		
	**CM**	**SfM**	**fCM**	**CM**	**SfM**	**fCM**	**CM**	**SfM**	**fCM**
	**[μm]**	**[μm]**	**[μm]**	**[mm${}^{-2}$]**	**[mm${}^{-2}$]**	**[mm${}^{-2}$]**	**[μm]**	**[μm]**	**[μm]**
P40	411	162	190	1.4	0.59	0.56	138.2	64.1	75.2
P60	250	78	83	3.5	0.88	0.84	86.7	32.0	35.1
P80	160	54	47	6.9	1.04	1.09	61.0	25.0	20.9
P100	132	34	32	9.7	1.08	1.26	51.4	18.5	16.6
P120	99	32	24	16.2	1.16	1.28	40.1	16.7	11.4
P180	62	28	10	32.4	1.23	2.92	25.2	16.3	7.8
P240	45	22	8	58.5	1.04	4.83	19.9	14.1	7.1

**Table 6 sensors-23-00178-t006:** Parameters of sandpaper P40 artifact from SfM measurements at a varying range and fCM data of a varying filter width. Values shown are the average particle height, particle area density as well as the $S_{q}$ roughness parameter.

Range	Av. Height		Density		Sq	
	**SfM**	**fCM**	**SfM**	**fCM**	**SfM**	**fCM**
	**[μm]**	**[μm]**	**[mm${}^{-2}$]**	**[mm${}^{-2}$]**	**[μm]**	**[μm]**
1.5 m	172	207	0.64	0.91	67.0	81.9
1.7 m	166	195	0.60	0.54	66.5	75.2
2.0 m	138	164	0.57	0.50	54.6	66.1
2.2 m	109	150	0.50	0.45	46.2	60.9

## Data Availability

Structure from Motion images, microscopy ground truth reconstructions and STL files of the used artifacts can be found in the dataset “Sandpaper Wind Turbine Blade Benchmark Dataset” [69].

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
