# Peer review of "Quantifying the Influence of Surface Texture and Shape on Structure from Motion 3D Reconstructions"

_sensors, 2022, doi:10.3390/s23010178_

Round 1

Reviewer 1 Report

1. Research Target:

In kinematic structures, surface reconstruction is crucial. Still, the study of the effect of surface texture on geometric reconstruction is often neglected, and this paper is intended to remedy this deficiency.

2. Method:

The artifacts were designed with well-defined surface geometry, and quantitative metrics were introduced to assess the reconstructions' lateral resolution, longitudinal geometric variation, and spatial frequency information. For the SfM measurements, ContextCapture software and a 50Mpx DSLR camera were used. Introduced metrics allow quantitative comparison of reconstructions at different texture levels and ranges.

3. Result:

A certain degree of texture is required in SfM reconstructions, while surface texture also heavily influences the reconstructed surface topography of SfM DEMs. PSD analysis shows that noise increases at low roughness levels. For step height artifacts with projected light patterns and sandpaper artifacts with fine grain size, this was observed to have a more significant PSD intensity at high spatial frequencies

4. Shortcoming:

(1). The purpose of the study is not very clear;

(2). The conclusion is not very straightforward;

Author Response

The authors would like to thank the reviewer for their comments on the proposed manuscript. We have taken steps to address the shortcomings pointed out:

(1). The purpose of the study is not very clear;

To clarify the purpose of the proposed study, we have made the following changes to the paper:

  • Expanded the Introduction section with further comparisons to the state-of-the-art methods for 3D reconstruction and pointing out the necessity of quantifying the effects of surface textures and shapes on SfM (Lines 70-82)
  • Added a Contributions subsection to the Introduction where we clearly specify the contribution of the paper to the field (Lines 116 - 126)
  • Further expanded the Introduction with initial Hypotheses and goals we start from in our study (Lines 130 -146)

(2). The conclusion is not very straightforward;

Two main changes have been done to address this problem:

  • The Discussion section has been expanded with a more straightforward overview of the insights and results from the analysis session and how they would influence possible further surface capturing. (Lines 509 - 520)
  • Completely rewritten the Conclusion section to better outline the take-home messages from the analysis of both artifacts and comparisons in the influence of GSD to the quality of the reconstructions. As well as added better outlined future work (Lines 542 - 562)

Reviewer 2 Report

Overall, this paper is quite good with a well-organized and interesting topic.

The authors use enough experiments to verify and discuss the influence of surface texture and shape. Although with marginal novelty, I think it is a meaningful work.

Some minor issues:

1. ContextCapture is a closed-source commercial software, why use this method instead of the widely used algorithms? It seems this paper is like verifying the effectiveness of ContextCapture

2.  In fact, photometric stereo, as the shape recovery method, theoretically is not impacted by the surface texture but depends on the lighting. Therefore, photometric stereo can even reconstruct well on a non-textured white wall. I think the author should briefly introduce this kind of method in your background part, with some references, such as:

[1] Woodham R J. Photometric Method For Determining Surface Orientation From Multiple Images[J]. Optical Engineering, 1980, 19(1): 191139.

[2] Ju Y, Shi B, Jian M, et al. NormAttention-PSN: A High-frequency Region Enhanced Photometric Stereo Network with Normalized Attention[J]. International Journal of Computer Vision, 2022, 130(12): 3014-3034.

[3] Wu L, Ganesh A, Shi B, et al. Robust photometric stereo via low-rank matrix completion and recovery[C]//Asian Conference on Computer Vision. Springer, Berlin, Heidelberg, 2010: 703-717.

Author Response

The authors would like to thank the reviewer for their insightful comments and the problems that they caught. We believe that the paper's readability and flow have been greatly improved.

The following changes have been made to address the reviewer's comments:

  1. ContextCapture is a closed-source commercial software, why use this method instead of the widely used algorithms? It seems this paper is like verifying the effectiveness of ContextCapture

The authors agree with the reviewer's concerns. ContextCapture is a fairly high-end and expensive program, but the authors wanted this paper to give the best possible "chance" for SfM reconstruction. This was done by using a top-of-the-line DSLR camera and SfM software solution that are used in indoor closely monitored conditions. This way we can demonstrate the best possible results that we could achieve and that could be used as a baseline for comparison to other researchers, together with the easy-to-reproduce artifacts.

We further address this by adding a number of additions to the paper:

  •  Overview of the paper testing Methodology and our desire for this paper to be used as a comparison/baseline. (Lines 17-19) (Lines 152 - 161) (Lines 33-37). As well as new overview of our hypotheses going into this research  (136-146)
  • Adding additional references to the performance of ContextCapture and why we choose to use it (LInes 211-218)

2. In fact, photometric stereo, as the shape recovery method, theoretically is not impacted by the surface texture but depends on the lighting. Therefore, photometric stereo can even reconstruct well on a non-textured white wall. I think the author should briefly introduce this kind of method in your background part, with some references, such as:

We agree with the reviewer's suggestion and proposed references. By introducing photometric stereo we can show a technology that does not suffer from the same problems as SfM and compare them together, giving a better understanding when and how SfM should be used.

We have integrated the suggestion by expanding the Introduction section and taking a deeper look at photometric stereo and comparing it to SfM and photogrammetry (Lines 70 -82). We have also integrated the proposed references, as well as adding additional ones  (Lines 27-28) (Lines 106-112)

Reviewer 3 Report

This study tries to remedy that by investigating the influence of object texture levels on reconstruction accuracy using a set of reference artifacts. This paper is well written and organized. However, I have the following concerns:

1 The contribution is unclear. In the part of introduction, it is better to list the contributions with in-depth analysis.

2 In 3D reconstruction, surface texture and shape are important factors to capture the reconstruction. I notice there are several references that investigate the effect of surface. The authors should analyse the literature again and provide the novelty of the proposed method, such as [1], [2]

[1] Quantifying the Effect of River Ice Surface Roughness on Sentinel-1 SAR Backscatter

[2] 3D optical surface profiler for quantifying leaf surface roughness 

3 There are extensive experimental analysis in this paper, which is welcome. However, the reviewer wants to see more results compared with related baselines.

4 Its better to provide a Conclusion section.

Author Response

First off, the authors would like to thank the reviewer for their in-depth comments and suggestions. We have taken them seriously and believe that the changes we have made to the manuscript based on them have resulted in greater readability overall and expanding further the state-of-the-art review sections, as well as the conclusions.

We will now give all the changes made based on each of the comments of the reviewer.

1 The contribution is unclear. In the part of introduction, it is better to list the contributions with in-depth analysis.

We absolutely agree with this point. We have addressed this by expanding the Introduction section greatly to contain a Contributions sub-section, as well as an in-depth look on all our starting Hypotheses for planning and organizing the methodological testing pipeline. We have given a more in-depth analysis and have reiterated our desire to make the paper a baseline example of comparison between SfM and microscopy 3D reconstruction that can be repeated with other types of SfM solutions or other methods for 3D reconstruction. (Lines 116 -126) (Line 128 - 146)

2 In 3D reconstruction, surface texture and shape are important factors to capture the reconstruction. I notice there are several references that investigate the effect of surface. The authors should analyse the literature again and provide the novelty of the proposed method, such as [1], [2]

We agree with this and have both added the proposed references, as well as adding additional dedicated references for the problems of determining the quality of the surface analysis methods. (Line 60) (Lines 93-95) (Lines 106-112) In addition we have more closely focused on references demonstrating the work of others in different use case requiring surface reconstruction and capturing details (Line 27)

3 There are extensive experimental analysis in this paper, which is welcome. However, the reviewer wants to see more results compared with related baselines.

4 Its better to provide a Conclusion section.

To better tie our experiments and contributions to related research we have rewritten the Conclusion and Discussion sections of the paper. We have added further discussion of our results in the context of knowledge from related baselines, as well as giving a better overview of what are the most important take-home messages from all the comparisons between the microscopy and SfM reconstructions. (Lines 358-360) (Lines 609 -619) (Lines 642-662)

Round 2

Reviewer 1 Report

My opinions of the improvement section:

(1). A comparative analysis was added to the introduction;

(2). The research methods are clearly defined;

(3). They have already added a conclusion summary to make the article's research results more transparent.